# A Deep Learning-Based Model for Predicting Abnormal Liver Function in Workers in the Automotive Manufacturing Industry: A Cross-Sectional Survey in Chongqing, China

**DOI:** 10.3390/ijerph192114300

**Published:** 2022-11-01

**Authors:** Linghao Ni, Fengqiong Chen, Ruihong Ran, Xiaoping Li, Nan Jin, Huadong Zhang, Bin Peng

**Affiliations:** 1School of Public Health, Chongqing Medical University, Chongqing 400016, China; 2Department of Occupational Health and Radiation Health, Chongqing Center for Disease Control and Prevention, Chongqing 400042, China

**Keywords:** abnormal liver function, deep learning, automotive manufacturing industry, risk factors, predictive model

## Abstract

To identify the influencing factors and develop a predictive model for the risk of abnormal liver function in the automotive manufacturing industry works in Chongqing. Automotive manufacturing workers in Chongqing city surveyed during 2019–2021 were used as the study subjects. Logistic regression analysis was used to identify the influencing factors of abnormal liver function. A restricted cubic spline model was used to further explore the influence of the length of service. Finally, a deep neural network-based model for predicting the risk of abnormal liver function among workers was developed. Of all 6087 study subjects, a total of 1018 (16.7%) cases were detected with abnormal liver function. Increased BMI, length of service, DBP, SBP, and being male were independent risk factors for abnormal liver function. The risk of abnormal liver function rises sharply with increasing length of service below 10 years. AUC values of the model were 0.764 (95% CI: 0.746–0.783) and 0.756 (95% CI: 0.727–0.786) in the training and test sets, respectively. The other four evaluation indices of the DNN model also achieved good values.

## 1. Introduction

The liver is an important solid organ in the body and a key hub for important physiological processes, such as nutrient metabolism, regulation of immune function, blood volume control, endocrine regulation, and lipid and cholesterol regulation [1]. Approximately 2 million people die from liver disease globally each year, making it a critical issue for global health today [2]. In China, liver disease affects approximately 300 million people. Because accurate statistics are not available for many areas where liver disease is highly prevalent (e.g., Africa), the health risks of liver disease to the world’s population are actually much worse than we currently know [3]. Occupational injury is one of the main causes of impaired liver function [4]. Numerous studies have found that multiple occupations can increase a worker’s risk of developing abnormal liver function. Somayeh et al. [5] found significantly higher levels of alanine aminotransferase (ALT) and aspartate aminotransferase (AST) in the blood of gas station workers than in controls; Deng et al. [6] found that occupational manganese exposure interacted with alcohol consumption to significantly exacerbate elevated liver enzyme concentrations; Steven et al. [7] showed that cab drivers working for long periods of time had significantly elevated ALT in their blood.

From 33 million units in 1975 to 73 million units in 2007, the global annual production of automobiles has almost doubled. The automotive manufacturing industry has grown rapidly and is now one of the world’s largest and most important industries [8]. In 2002 and 2003, China’s automobile production increased by 37.1% and 35.1%, respectively, and in 2010, the auto industry became a pillar industry in China [9]. Workers in the automotive manufacturing industry are mainly responsible for the production and processing of auto parts and trims, and the manufacture of complete automobiles. The process of automobile production involves mechanical, electronic, chemical, and other industries, and there are dust, chemical toxins, noise, vibration, high temperature, human ergonomics, occupational stress, and other occupational harmful factors. Workers suffer from a wide range of occupational harmful factors, which can cause a variety of occupational injuries. Ana et al. [10] found in a study of 228 assembly and paint workers in an automotive manufacturing company that the occupational environment they were exposed to significantly increased their risk of musculoskeletal pain. Luo et al. [11] found that automotive manufacturing workers were at greater risk of hearing loss than the exposed group and that the longer the actual working age of exposure to noise, the greater the probability of hearing loss and the more severe the hearing loss. Andre et al. [12] studied 75 shift workers in the automotive manufacturing industry and found that they had a significantly higher risk of cardiovascular disease than the control group population. In addition to the damage to the musculoskeletal, hearing, and cardiovascular systems, the occupational environment of workers in the automotive industry can also cause serious damage to their liver function. Previous studies have shown that organic solvents, volatile organic compounds, extremely low-frequency electromagnetic fields, and high temperatures in the automotive industry can cause liver damage [13,14,15,16]. The current outlook of abnormal liver function among workers in the automotive manufacturing industry is not optimistic. There is an urgent need to conduct a comprehensive analysis of the workers’ working environment and identify the risk factors in the environment that endanger workers’ liver function. A timely, effective, and accurate early warning system is also crucial for industrial environment rectification, policy formulation, and individualized worker management.

Alanine aminotransferase (ALT) is an aminotransferase enzyme found in blood plasma and various body tissues but commonly found in the liver [17], and is the primary aminotransferase used in clinical blood tests for liver function. As little as 1% of liver cell damage can increase the concentration of ALT in the blood by a factor of 1. Therefore, the ALT level can be a sensitive measure of whether the liver is damaged. The World Health Organization recommends ALT as the most sensitive indicator of liver function damage, which is now widely used as a marker of hepatobiliary disease in occupational health monitoring. In this study, the blood ALT level was used as the evaluation criterion for abnormal liver function, and an ALT level of 0–40 IU/L indicates normal liver function, while an ALT level greater than 40 IU/L indicates abnormal liver function [18].

Traditional studies of influencing factors can provide ideas for the control of disease epidemics. However, in the automotive manufacturing industry, we are more concerned about the risk of worker morbidity, issuing timely and accurate warnings, and taking appropriate measures to ensure the personal safety of workers, which is a problem that cannot be solved by traditional research. Predictive models using machine learning and deep learning are widely used today. Many studies have developed models to predict abnormal liver function or liver disease. Shaker Abdalrada et al. [19] developed a logistic regression model using laboratory tests to predict liver disease. Yip et al. [20] used clinical data and laboratory data to develop predictive models for non-alcoholic fatty liver disease using four machine learning algorithms. Ma et al. [21] developed a predictive model for the diagnosis of non-alcoholic fatty liver disease based on an integrated machine learning method. Jiang et al. [22] used the BP neural network to predict chronic liver disease. Xu et al. [23] conducted a cross-sectional study to predict non-alcoholic fatty liver disease by using the Bayesian network model. Although there have been many studies that have developed predictive models related to liver disease, they have largely used laboratory data as the independent variable. In our practical applications, it is often difficult to obtain laboratory data, which reduces the convenience of the model as a tool. Most current studies focus on patients or the general population. Although the health of workers in the automotive manufacturing industry has been a hot topic of research, there is a paucity of research on predictive models of liver function in workers (Table 1).

There is a need to address the two main limitations in current studies: the lack of convenient models to use and the lack of research on automotive manufacturing workers as a population. In this study, our main objective was to develop a predictive model of liver function. This model relies only on demographic characteristics and occupational information. It will not only fill the gap of related studies but also serve as a convenient tool for workers’ health management. This is the main innovation and contribution of our study. In addition to this, as a cross-sectional study, our study also revealed the current prevalence of abnormal liver function and its influencing factors among automotive manufacturing workers in Chongqing.

The remaining sections are organized as follows. We introduce the materials and methods in Section 2 which specifically covers the study subjects (Section 2.1), data collection (Section 2.2), and statistical analysis (Section 2.3). Then we show the results of our study in Section 3 which specifically covers descriptive analysis (Section 3.1), characteristics comparison (Section 3.2), identification of risk factors (Section 3.3), the impact of length of service (Section 3.4), and the performance of the model (Section 3.5). Finally, we discuss our results and conclude our study in Section 4 and Section 5, respectively.

## 2. Materials and Methods

### 2.1. Study Subjects

The subjects of this study were 16,384 workers in the automotive manufacturing industry in Chongqing from 2019 to 2021. Data without information on ALT, demographic characteristics, and work environment information (*n* = 9735), and data with missing values or logical errors (*n* = 562) were removed according to the requirements of this study, resulting in the inclusion of 6087 individuals in the study (Figure 1). The data were obtained from occupational health inspections and workplace testing by institutions qualified to conduct occupational health inspection and workplace testing in accordance with the relevant state regulations.

### 2.2. Data Collection

All case physical examination data for workers were obtained from occupational health examinations, which are operated according to the Technical Specification for Occupational Health Surveillance [S] GBZ 188-2014 and are conducted in a private environment. Checks for consistency were performed before, during, and after data entry, and strict confidentiality was maintained regarding all worker identification information and medical examination data. Other necessary information was obtained through questioning and corporate records, resulting in clinical and work information, including age, sex, height, weight, diastolic blood pressure (DBP), systolic blood pressure (SBP), length of service, and scale of enterprise. Body mass index (BMI) was calculated as weight (kg)/height (m)^2^. Blood pressure measurement: blood pressure values of the study subjects were measured using the Omron arm blood pressure monitor. Blood pressure was measured three times on the same subject, each time more than two hours apart, and the final blood pressure value was the average of the three measurements. Noise exposure detection: individual noise dose meter was selected, and the measurement instrument was calibrated according to the instrument calibration requirements and then used for measurements; the microphone should point in the direction of the sound source and be placed at the ear of the worker at work, 1.50 m in a standing position and 1.10 m in a sitting position; the detection point is the workplace of steady-state noise, each detection point is measured three times, and the average value is taken. The standard for excess noise exposure is an 8-h equivalent A sound level greater than 85 dB. The specific testing operation procedure refers to *GBZ/T 189.8*, which stipulates the method of measurement. Benzene exposure detection: the performance and specifications of the air collector and air sampler used must be checked before sampling, and the sampling flow rate and timing device of the air sampler must be corrected before collection. The testing operation procedure refers to the Determination of Air Toxic Substances in the Workplace Part 66: Benzene, Toluene, Xylene GBZ/T 300.66-2017 specified method measurement. Abnormal liver function according to the National Clinical Test Procedure 4th Edition: ALT in the range of 0–40 IU/L indicates normal liver function, and ALT greater than 40 IU/L indicates abnormal liver function.

### 2.3. Statistical Analysis

#### 2.3.1. Identification of Influencing Factors

Continuous variables were described by mean ± standard deviation, and categorical variables were described by frequency and percentage. Differences between groups were compared using a t-test or chi-square test. Logistic regression analysis is a non-linear regression, which is a multiple regression analysis method to study the relationship between the dependent variable as a dichotomous or multinomial classification result and some influencing factors. In this study, the conditional probability of occurrence of abnormal liver function under the action of the independent variable can be expressed as:(1)$$P=\frac{\exp\left(\beta_{0}+\beta_{1}x_{1}+\beta_{2}x_{2}+\cdots +\beta_{m}x_{m}\right)}{1+\exp\left(\beta_{0}+\beta_{1}x_{1}+\beta_{2}x_{2}+\cdots +\beta_{m}x_{m}\right)}$$
where: $\beta_{0}$ is the constant term, $\beta_{1}$, $\beta_{2}$, …, $\beta_{m}$ is the partial correlation coefficient. Univariate logistic regression was used for the analysis of influencing factors, and the odds ratio (OR) and its 95% confidence interval (CI) were calculated. Variables with univariate logistic regression analysis *p*-values less than 0.05 were further analyzed using multivariate logistic regression analysis to identify independent influences.

#### 2.3.2. Restricted Cubic Spline Analysis

The length of service data of workers is readily available and have a significant effect on workers’ health. The relationship between the length of service and the risk of abnormal liver function is not a simple linear relationship, and logistic analysis cannot fully reveal it. A restricted cubic spline model was used to further explore the nonlinear relationship between the length of service and the risk of abnormal liver function. Let the range of the independent variable data be in the interval [a, b], and divided into *k* segments as needed: $a=t_{0}<t_{1}<\cdots <t_{k-1}<t_{k}=b$. Each interval [$t_{i-1}$, $t_{i}$) is represented by a polynomial equation $S_{i}\left(x\right)$, respectively. Then, the spline function can be expressed as:(2)$$RCS\left(x,k\right)=\overset{k-1}{\underset{i=1}{\sum }}\beta_{i}S_{i}\left(x\right)$$
where: k is the number of intervals to be divided; $\beta_{i}$ is the partial correlation coefficient; $S_{i}\left(x\right)$ is the polynomial equation in each interval.

All statistical analyses in this study were conducted using R software (version 4.1.2) (R Core Team, Vienna, Austria). All statistical tests were performed using a two-sided test with a level of α = 0.05.

### 2.4. Development and Evaluation of DNN Model

The DNN was established with abnormal liver function as the dependent variable and variables with *p*-values less than 0.05 in the univariate logistic regression as the independent variables. In this study we used a fully-connected neural network architecture to train our DNN model. For our model, the output of the level *n* can be expressed as:(3)$$A^{n+1}=\delta^{n}\left(W^{n}A^{n}+b^{n}\right)$$
where $W^{n}$ is the weights of level *n*; $b^{n}$ is the bias of level *n*; $\delta^{n}$ is the activation of level *n*. Since the DNN requires the variables to take values between 0 and 1, the model was built by performing the $\frac{x-x_{min}}{x_{max}-x_{min}}$ transformation on the continuous variables. Some samples are shown in Table 2. Y indicates abnormal liver function as the dependent variable and *x*1–*x*9 indicates the independent variables (age, BMI, length of service, DBP, SBP, sex, benzene exposure, noise exposure, and size of enterprise). To evaluate the performance of the DNN model, we further established the logistic regression model (LR), the eXtreme Gradient Boosting model (XGBoost), and the support vector machine model (SVM). AUC, accuracy, sensitivity, specificity, and F1-score were used to compare the DNN model with these three models. We first divided all samples into a train set and a test set in the ratio of 7:3. The train set is used to train the model and for parameter tuning, and the test set is used to evaluate the generalization ability of the models. No parameter tuning is required for the LR model. In the train set, we used a grid search in order to perform parameter tuning while training the three models. A 10-fold cross-validation method (in the train set) was used to select the optimal parameters. After selecting the optimal parameters, the final models to be used are determined, and they are used to make predictions on the test set data and to derive the final evaluation metrics. The receiver operator characteristic curves (ROC curves) were plotted and the value at the maximum of the Jorden index was taken as the cutoff of the model prediction results. The trained models were used to predict the data in the test set and obtain the performance of the models.

## 3. Results

### 3.1. Demographic Characteristics and Work Environment Information of Workers

A total of 6087 automotive manufacturing workers were surveyed in this study. The average age of the 6087 workers was 36.8 ± 10.5 (years); the average BMI was 23.5 ± 3.5 (kg/m^2^); the average DBP was 80.6 ± 10.9 (mmHg); the average SBP was 125.0 ± 15.3 (mmHg); the average length of service was 6.9 ± 7.05 (years); there were 5178 male workers, accounting for 85.1% of the study subjects; 1117 benzene-exposed workers, accounting for 18.4%; 3884 noise-exposed workers, accounting for 63.8%; 929 (15.3%) cases of workers in small enterprises, 1189 cases (19.5%) in medium-sized enterprises, and 3969 cases (65.2%) in large enterprises (Table 3).

### 3.2. Comparison of Characteristics between the Normal Liver Function Group and Abnormal Liver Function Group

Of all 6087 study subjects, a total of 1018 cases (16.7%) were detected with abnormal liver function. Compared to the normal liver function group, workers in the abnormal liver function group had higher BMI (25.6 ± 3.39 vs. 23.1 ± 3.32), length of service (7.59 ± 7.09 vs. 6.80 ± 7.03), DBP (84.2 ± 11.6 vs. 79.9 ± 10.6), and SBP (130 ± 15.7 vs. 124 ± 15.0) than the normal liver function group; while age (35.4 ± 9.39 vs. 37.0 ± 10.7) was lower than that of the normal liver function group. The noise exposure rate (66.9% vs. 63.2%) and male rate (94.9% vs. 83.1%) were higher in the abnormal liver function group compared to the normal liver function group, and the distribution of enterprise size was different between the two groups (*p* < 0.05) (Table 4).

### 3.3. Identification of Risk Factors for Abnormal Liver Function

A univariate logistic regression analysis with abnormal liver function as the dependent variable and age, BMI, length of service, DBP, SBP, sex, benzene exposure, noise exposure, and size of enterprise as independent variables found that: Compared to female workers, male workers are 3.78 times (OR = 3.78; 95% CI: 2.832–5.044) more likely to have abnormal liver function. For each additional year of length of service, workers are 0.015 times (OR = 1.015; 95% CI: 1.006–1.024) more likely to develop abnormal liver function. Besides, increases in BMI (OR = 1.232; 95% CI: 1.207–1.258), DBP (OR = 1.036; 95% CI: 1.03–1.042), and SBP (OR = 1.025; 95% CI: 1.021–1.029), and noise exposure (OR = 1.177; 95% CI: 1.021–1.358) and working in a large enterprise (OR = 1.551; 95% CI: 1.256–1.914) were all risk factors for abnormal liver function. While for each additional year of age, workers are 0.015 times (OR = 0.985; 95% CI: 0.979–0.992) less likely to have liver function abnormalities. Multivariate logistic regression analysis revealed that BMI (OR = 1.218; 95% CI: 1.192–1.244), length of service (OR = 1.022; 95% CI: 1.010–1.034), DBP (OR = 1.017; 95% CI: 1.007–1.027), SBP (OR = 1.008; 95% CI: 1.000–1.015), and being male (OR = 3.272; 95% CI: 2.418–4.428) were all independent risk factors for abnormal liver function (Table 5).

### 3.4. The Relationship between Length of Service and Risk of Abnormal Liver Function

Based on restricted cubic spline regression analysis, we further investigated the dose–effect relationship between length of service and the risk of abnormal liver function in workers. The results showed a significant non-linear relationship between length of service and the risk of abnormal liver function: the OR rose sharply with length of service when it was less than 10 years; after 10 years, the OR leveled off and remained at about 2 (Figure 2).

### 3.5. A predictive Model for Abnormal Liver Function in Workers in the Automotive Manufacturing Industry

The study subjects were divided into a training set (*n* = 4261) and a test set (*n* = 1826) according to a ratio of 7:3. Abnormal liver function was used as the dependent variable, and age, BMI, length of service, DBP, SBP, sex, noise exposure, and size of the enterprise were used as independent variables according to the inclusion criteria of univariate logistic regression analysis with a *p*-value less than 0.05. Because of the extensive literature documenting that benzene exposure is significantly associated with liver function, benzene exposure was also included in the independent variables of the model in this study, resulting in the inclusion of nine independent variables. The optimal parameters of the three models were derived using the train set (Appendix A). DNN was built using the R package “neuralnet” (version 1.44.2), and the final model hyperparameters are set as follows: the model contains three hidden layers, and the number of neurons in each hidden layer is three. The backpropagation algorithm is used, and “logistic” is chosen as the activation function. After one epoch, the model training is completed (Figure 3A). The model achieved good predictions in both the training set (AUC = 0.764; 95% CI: 0.746–0.783) and the test set (AUC = 0.756; 95% CI: 0.727–0.786) (Figure 3B,C). Based on the ROC curve of the training set, the final prediction cutoff was determined to be 0.161 according to the principle of maximizing the Jorden index. In the train set, the AUC (0.774), accuracy (0.693), specificity (0.687), and F1-score (0.442) of the XGBoost model are all the highest among the four models, while the sensitivity (0.734) of the LR model is the highest. The AUC (0.764), accuracy (0.689), sensitivity (0.713), specificity (0.684), and F1-score (0.432) of the DNN model are ranked third, second, fourth, second, and second, respectively, among the four models (Figure 4A). In the test set, the AUC (0.756), accuracy (0.684), and F1-score (0.413) of DNN model are all the highest among the four models, and the sensitivity (0.712) and specificity (0.679) are ranked second, while the sensitivity (0.731) of the LR model and the specificity (0.689) of the XGBoost model are the highest among the four models (Figure 4B).

## 4. Discussion

This study investigated the current prevalence of abnormal liver function among workers in the automobile manufacturing industry in Chongqing. We comprehensively analyzed the effects of worker demographic characteristics and work environment factors on abnormal liver function. It was found that the prevalence of abnormal liver function among workers in the automobile manufacturing industry in Chongqing was 16.7%, which was significantly higher than the prevalence of liver diseases in the total population in most regions of China (2.3–6.1%) [24]. The occupational abnormal liver function ouotlook in the automotive manufacturing industry is not optimistic.

In the present study, we find that age was a protective factor for abnormal liver function (OR = 0.969; 95% CI: 0.960–0.978), which is inconsistent with the results of previous studies that considered age as a risk factor [25]. The possible reason for this is that although some studies point to a decrease in liver volume and blood flow and a decrease in hepatobiliary function with increasing age, a significant decline occurs after old age (>60 years) [26,27]. In contrast, the subjects of the present study were employed workers in the automotive manufacturing industry, most of whom were in their prime (36.8 ± 10.5) and had a very slight diminution of liver function due to age. Together with the influence of other complex factors, it may be the main reason for the conclusions of this study. Increased BMI is also a significant factor in abnormal liver function (OR = 1.218, 95% CI: 1.192–1.244). Studies have shown that obesity accelerates epigenetic aging of the liver, with an average increase of 3.3 years in epigenetic age of the liver for every 10 BMI units [28]; at the same time obesity reduces the role of the liver in lipid metabolism and even causes steatosis of liver cells. This series of changes significantly increases the risk of fatty liver and nonalcoholic hepatitis [29,30]. Certain jobs in the automotive manufacturing industry (such as packers) require long periods of sitting and lack of necessary activity, which can greatly increase their risk of obesity and thus increase their vulnerability to abnormal liver function. The management of enterprises should pay more attention to the weight detection of this group of works, timely detection, and taking appropriate measures. At the same time, employees should be actively organized to participate in sports activities and promote exercise to reduce the risk of obesity among employees. In addition, increased DBP (OR = 1.017; 95% CI: 1.007–1.027) and SBP (OR = 1.008; 95% CI: 1.008) in workers were both independent risk factors for abnormal liver function. This is consistent with the conclusion that hypertension is capable of causing abnormal increases in serum liver enzymes in previous studies [31]. Long-term hypertension is very likely to cause arteriosclerosis, and if arteriosclerosis occurs in the liver vessels, it will cause insufficient blood supply to the liver and even affect liver function [32]. If the worker’s blood pressure has reached the point where he must rely on drugs to lower it, long-term use of drugs can also increase the burden on the liver and affect liver function. Many workers in the automotive manufacturing industry are exposed to factors such as heat, noise, and organic compounds that can cause hypertension, making strict blood pressure management even more important. Compared to females, male workers have a substantially increased risk of abnormal liver function (OR = 3.272; 95% CI: 2.418–4.428). This is because, in the automotive manufacturing industry, male workers are generally engaged in welding, forging, and other positions that are easily exposed to harmful factors, such as welding fumes and toxic volatile gases. Female workers, on the other hand, are mainly engaged in cleaning and other work, with less exposure to the corresponding hazardous factors. At the same time, a larger proportion of male workers drink alcohol, and alcohol is also an important cause of reduced liver function [33].

Length of service is also a significant risk factor for abnormal liver function (OR = 1.022; 95% CI: 1.010–1.034), and the information is accurate and easily available, which can provide an important basis for policymaking and health management in the automotive manufacturing industry. We found that increased BMI, length of service, DBP, SBP, and being male are independent risk factors for abnormal liver function. These findings may be more like common sense for us. To get a deeper understanding, in the current study, a restricted cubic spline model was used to further reveal the nonlinear dose–response relationship between the length of service and the risk of abnormal liver function. The OR rises sharply during the decade when workers begin working in automotive manufacturing. This may be due to sudden exposure to many harmful factors and the body’s inability to compensate in time resulting in an increased risk of abnormal liver function [34]. This finding suggests that we should pay more attention to the health management of new employees, and try to assign new employees to jobs with relatively few risk factors when they first join the company, and gradually adjust their positions. This does not mean that older employees are not cared for. The OR of older employees with more than 10 years of service remains at a high level, although their risk of disease does not rise significantly. For older employees, we should strengthen their health testing and organize regular health checkups.

Timely information on workers’ health status and understanding their risk of abnormal liver function is important for developing individualized management plans. In this study, a risk prediction model for abnormal liver function in workers in the automotive manufacturing industry was developed based on a deep neural network, and good predictions were achieved in both the training set (AUC = 0.764; 95% CI: 0.746–0.783) and the test set (AUC = 0.756; 95% CI: 0.727–0.786). Although the performance of the XGBoost model in the train set is the best (four indices ranked first and one index second) among the four models, its performance in the test set is average. This indicates that the XGBoost model produces slight overfitting and is not suitable for practical application. For feedforward neural networks, the depth of the credit assignment path (CAP) is the depth of the network, which is the number of hidden layers plus one. There is no universally accepted depth threshold that distinguishes shallow learning from deep learning, but most researchers agree that deep learning involves a CAP depth higher than 2 [35]. Since our network is a feed-forward neural network and has three hidden layers (CAP depth of four), we refer to our neural network as deep learning. We have also tried to increase the number of hidden layers of our neural network to make it deeper in our research, but we found that overfitting occurs at more than three hidden layers in our data. We finally chose to build our neural network with only three hidden layers. Among the four models, the DNN model performs second (three indices ranked second, one index third, and one index fourth) only to the XGBoost model in the training set, and is the best (three indices ranked first and two indices second) performing model in the test set. The performance of a model in a test set is often representative of its performance in real-world applications. In addition, the performance of the DNN model is very close in the train and test sets, which indicates that the DNN model has no overfitting and has good generalization ability. Although the LR model has the highest sensitivity in both the train and test sets, its other indices perform mediocrely. A performance with high sensitivity and low specificity means that the LR model has a high probability of causing a “misdiagnosis”. The performance of the SVM model is inferior to the other three models in all aspects. Collectively, the DNN model has the best results among the four models established in this study. In addition, compared to other medical-related modeling studies, our study achieves superior predictive results (such as the AUC of 0.764 in this study vs. the AUC of 0.740 in Wang [36]). In summary, our established DNN model not only has good prediction performance, but also has strong generalization ability. In addition to that, the DNN model was built based on simple and easily available information (demographic characteristics and occupational information). With this model, company managers can keep track of the liver function of workers in the automotive industry without tedious laboratory tests. It can be applied to the practical application of the development of management policies, for example, adjusting the working hours and jobs of workers. Combining the use of the DNN model, regular worker health checkups, and work environment quality testing can more effectively protect the health of workers.

### Study Limitations and Future Works

Personal lifestyle (diet, consumption of alcohol, physical activity level, etc.) also has an important influence on liver function. However, our study did not address the relevant aspects. The current study was conducted on auto manufacturing workers in Chongqing, and the performance of the model in populations around the world is unknown. In the future, we will try to collect more rich data in terms of region and content to verify and improve our research. This is a long and complex job. We have uploaded the main codes and the trained models for this study to github (https://github.com/little2b/A-deep-learning-based-model-for-predicting-abnormal-liver-function-in-workers-in-the-automotive-manu, accessed on: 22 October 2022) for those who need them. We also hope that researchers from around the world will use similar data to further evaluate the effects of our model on different populations. In addition to that, we will learn more and train more and better models.

## 5. Conclusions

Increases in BMI, length of service, DBP, SBP, and being male are independent risk factors for abnormal liver function. The risk of abnormal liver function in workers increases sharply with the increasing length of service below 10 years. The model based on the deep neural network has good performance and can be used as an effective tool to predict workers suffering from abnormal liver function, and provide personalized management plans.

## Figures and Tables

**Figure 1 ijerph-19-14300-f001:**
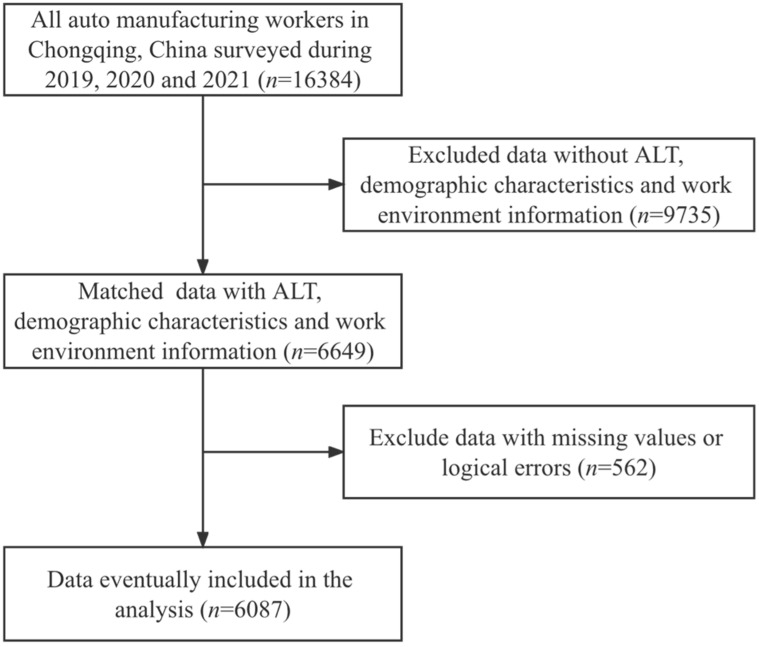
Flowchart of research subject enrollment.

**Figure 2 ijerph-19-14300-f002:**
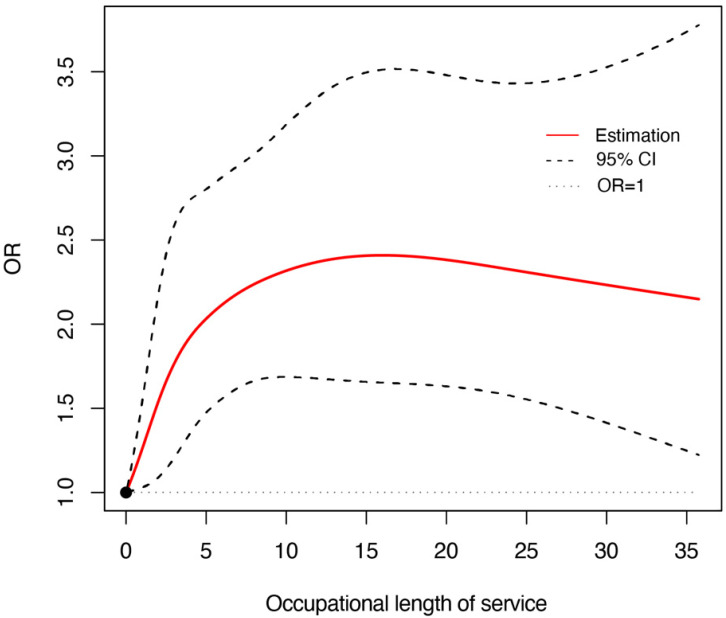
Dose–response relationship between the occupational length of service and the risk of abnormal liver function.

**Figure 3 ijerph-19-14300-f003:**
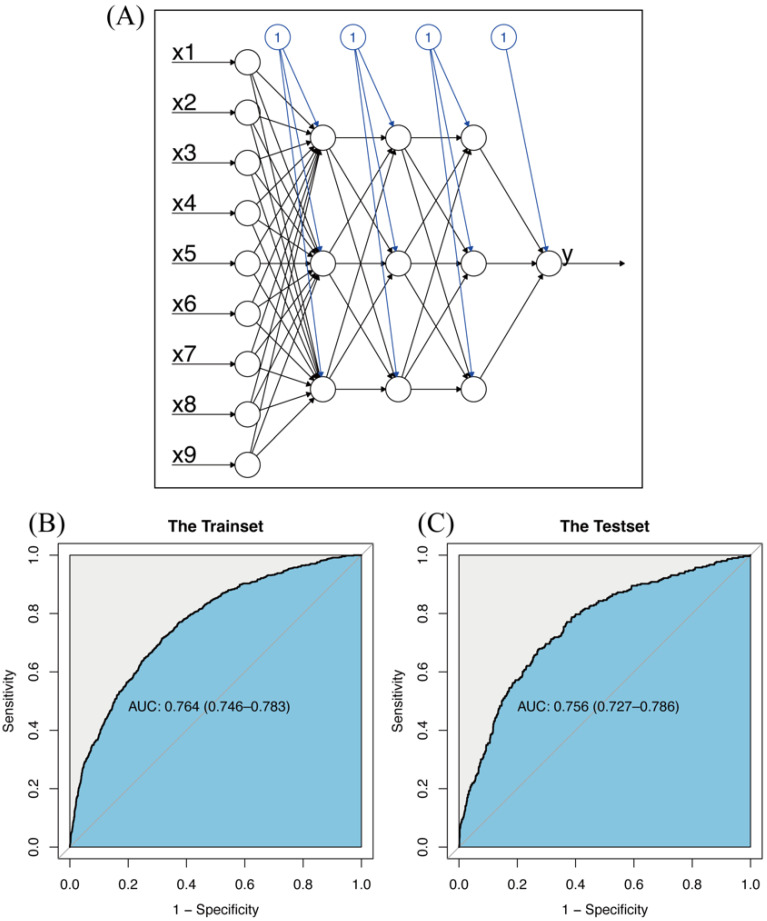
Predictive model for abnormal liver function (**A**) Deep neural network model structure diagram; y indicates abnormal liver function as the dependent variable; *x*1–*x*9 indicate the independent variables (age, BMI, length of service, DBP, SBP, sex, benzene exposure, noise exposure, and size of enterprise); the black circles in the figure represent the neurons in the deep neural network, and the blue circles represent the bias terms in the network; (**B**,**C**) ROC curves of the model in the training and test sets.

**Figure 4 ijerph-19-14300-f004:**
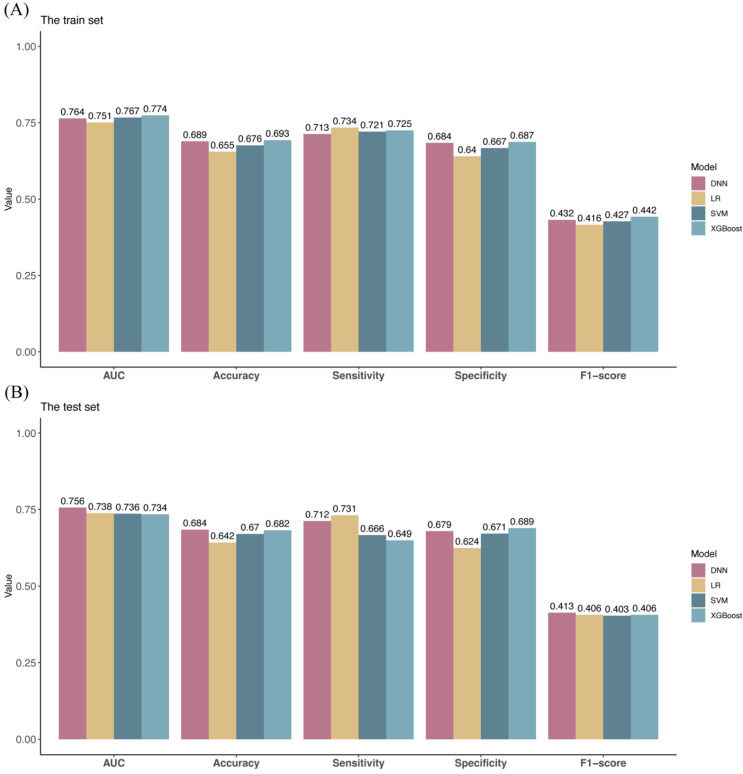
Performance of the four models (**A**) The performance of the four models in the train set. (**B**) The performance of the four models in the test set.

**Table 1 ijerph-19-14300-t001:** Reviews of the different models of liver-related disease.

	Study Subjects	Sample Size	Independent Variables	Model	Performance Index
Abdalrada [19]	Indian Liver Patient Dataset	583	Demographic characteristics and laboratory data	Logistic regression	AUC Accuracy Sensitivity Specificity
Yip [20]	Patients in hospital	922	Demographic characteristics and laboratory data	Logistic regression Ridge regression AdaBoost Decision tree	AUC Sensitivity Specificity
Ma [21]	Patients in hospital	98	Demographic characteristics and laboratory data	Logistic regression Random forest Support vector machine	AUC
Jiang [22]	Patients in hospital	35	Laboratory data	BP neutral network	AUC Accuracy MSE
Ma [23]	Patients in hospital	10,508	Demographic characteristics and laboratory data	Bayesian network	F-measure Accuracy Sensitivity Specificity Precision

**Table 2 ijerph-19-14300-t002:** Some samples of this study.

	*X*1	*X*2	*X*3	*X*4	*X*5	*X*6	*X*7	*X*8	*X*9	y
*S*1	1	0.139899	0.149886	1	0	0.136555	0.463415	0.34375	0	0
*S*2	0	0.313271	0.30896	0	1	0.132353	0.353659	0.242188	1	0
*S*3	1	0.450327	0.410568	0	0	0.058824	0.463415	0.382813	1	1
*S*4	0	0.499732	0.317005	1	1	0.428571	0.329268	0.40625	0	1

**Table 3 ijerph-19-14300-t003:** Demographic characteristics and work environment information of workers.

	Total (*n* = 6087)
Age (years)	
Mean ± SD	36.8 ± 10.5
BMI (kg/m^2^)	
Mean ± SD	23.5 ± 3.5
DBP (mmHg)	
Mean ± SD	80.6 ± 10.9
SBP (mmHg)	
Mean ± SD	125.0 ± 15.3
Length of service (years)	
Mean ± SD	6.9 ± 7.05
Sex	
Female	909 (14.9%)
Male	5178 (85.1%)
Exposure to benzene	
No	4970 (81.6%)
Yes	1117 (18.4%)
Exposure to noise	
No	2203 (36.2%)
Yes	3884 (63.8%)
Scale of enterprise	
Small and under	929 (15.3%)
Medium	1189 (19.5%)
Large	3969 (65.2%)

**Table 4 ijerph-19-14300-t004:** Comparison of characteristics between normal liver function group and abnormal liver function group.

	Normal Liver Function (*n* = 5069)	Abnormal Liver Function (*n* = 1018)	*p*-Value
Age (years)	37.0 ± 10.7	35.4 ± 9.39	<0.0001 ***
BMI (kg/cm^2^)	23.1 ± 3.32	25.6 ± 3.39	<0.0001 ***
Length of service (years)	6.80 ± 7.03	7.59 ± 7.09	0.0011 **
DBP (mmHg)	79.9 ± 10.6	84.2 ± 11.6	<0.0001 ***
SBP (mmHg)	124 ± 15.0	130 ± 15.7	<0.0001 ***
Sex (male)	4212 (83.1%)	966 (94.9%)	<0.0001 ***
Exposure to benzene	952 (18.8%)	165 (16.2%)	0.0587
Exposure to noise	3203 (63.2%)	681 (66.9%)	0.0271 *
Size of enterprise			<0.0001 ***
Small and under	813 (16.0%)	116 (11.4%)	
Medium	1006 (19.8%)	183 (18.0%)	
Large	3250 (64.1%)	719 (70.6%)	

* *p*-value < 0.05 ** *p*-value < 0.01 *** *p*-value < 0.001.

**Table 5 ijerph-19-14300-t005:** Univariate and multifactorial logistic regression analysis of factors influencing abnormal liver function.

		Univariate			Multivariate	
Variables	OR	95% CI	*p*-Value	OR	95% CI	*p*-Value
Age	0.985	0.979–0.992	<0.0001 ***	0.969	0.960–0.978	<0.0001 ***
BMI	1.232	1.207–1.258	<0.0001 ***	1.218	1.192–1.244	<0.0001 ***
Length of service	1.015	1.006–1.024	0.0011 **	1.022	1.010–1.034	0.0002 ***
DBP	1.036	1.03–1.042	<0.0001 ***	1.017	1.007–1.027	0.0011 **
SBP	1.025	1.021–1.029	<0.0001 ***	1.008	1.000–1.015	0.0393 *
Sex (vs. Female)	3.78	2.832–5.044	<0.0001 ***	3.272	2.418–4.428	<0.0001 ***
Exposure to benzene (vs. No)	0.837	0.698–1.002	0.0532	–	–	–
Exposure to noise (vs. No)	1.177	1.021–1.358	0.0248 *	1.142	0.974–1.340	0.1023
Size of enterprise (vs. Small and under)	–	–	–	–	–	–
Medium	1.275	0.993–1.638	0.0572	1.047	0.801–1.368	0.7375
Large	1.551	1.256–1.914	<0.0001 ***	1.11	0.874–1.408	0.3924

* *p*-value < 0.05 ** *p*-value < 0.01 *** *p*-value < 0.001.

## Data Availability

The datasets involved in the current study are not publicly available due to privacy but are available from the author Linghao Ni on reasonable request.

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
