# Peer review of "A Deep Learning-Based Model for Predicting Abnormal Liver Function in Workers in the Automotive Manufacturing Industry: A Cross-Sectional Survey in Chongqing, China"

_ijerph, 2022, doi:10.3390/ijerph192114300_

Round 1

Reviewer 1 Report

The paper approaches a topical issue presenting the results of the study on predicting abnormal liver functioning for workers in automotive manufacturing industry.

The cited references are relevant and many of them are published in the last 5 years.

Specific comments:

- Rows 117 - 137 Section 2.2. Data collection should be presented more detailed. Please, explain why the noise exposure was considered as a relevant factor for the study and introduce some information about the determined noise level.

- Section 4. Conclusion - Personal life style (alimentation, consuming of alcohol, physical activity level) has also an important influence on liver functioning and should be mentioned as a limitation, since it was not considered in the study.

Author Response

Point 1: Rows 117 - 137 Section 2.2. Data collection should be presented more detailed. Please, explain why the noise exposure was considered as a relevant factor for the study and introduce some information about the determined noise level.

Response 1: We thank you for raising this question. We reviewed some literature suggesting a correlation between noise exposure and abnormal liver function (Oliveira MJ, Freitas D, Carvalho AP, Guimarães L, Pinto A, Águas AP. Exposure to industrial wideband noise increases connective tissue in the rat liver. Noise and health. 2012 Sep 1;14(60):227. Khavanin A, Rezazade-Azari M, Vosooghi S. Exposure to noise pollution and its effect on oxidant and antioxidant parameters in blood and liver tissue of rat. Zahedan Journal of Research in Medical Sciences. 2013 May 31;15(5).). And more details and information about the data collection and noise were added to my new manuscript.

Point 2: Section 4. Conclusion - Personal life style (alimentation, consuming of alcohol, physical activity level) has also an important influence on liver functioning and should be mentioned as a limitation, since it was not considered in the study.

Response 2: We have added personal life style as a limitation to lines 404-406 of my new manuscript.

Reviewer 2 Report

The authors did a great investigation by utilizing a deep learning-based model for predicting abnormal liver function in workers in the automotive manufacturing industry:  a cross-sectional survey in Chongqing, China. However, adding the following points will boost the article's contribution:

1-Add a paragraph at the end of the Introduction to show how the rest of the sections are organized.

2-Increase your literature review by adding a table to compare between the very recent related works in terms of method used, performance metrics, dataset, ...etc

3-In materials and methods discuss the algorithm(s) used and clarify the mathematical equations (i.e.logistic regression, deep neural network learning).

4-In the Conclusion section outline the futuristics for your works.

I can accept the work after amendment,

Author Response

Point 1: Add a paragraph at the end of the Introduction to show how the rest of the sections are organized.  

Response 1: We thank you for raising this comment. We have added this paragraph to lines 114-120 of our new manuscript.

Point 2: Increase your literature review by adding a table to compare between the very recent related works in terms of method used, performance metrics, dataset, ...etc.

Response 2: We included relevant studies from the last five years in the introduction and summarized them in Table 1 of our new manuscript.

Point 3: In materials and methods discuss the algorithm(s) used and clarify the mathematical equations (i.e.logistic regression, deep neural network learning).

Response 3: We have described the basic mathematical formulation of the algorithm used in the Statistical analysis section of our new manuscript.

Point 4: In the Conclusion section outline the futuristics for your works.

Response 4: We have added the relevant content to lines 407 to 414 of our new manuscript.

Reviewer 3 Report

The authors presented a new technique for A deep learning-based model for predicting abnormal liver function in workers in the automotive manufacturing industry: cross-sectional survey in Chongqing, China. However, here are some minor observations:

      I.         Please separately add problem statement and contribution of the study in Introduction section.

    II.         Mention the challenges faced while implementation and how did you tackle these

  III.         Mention if there are any trade-offs for implementation of this approach while we deviate from traditional practices

  IV.         Some punctuation errors are observed. Please improve the punctuation in the manuscript.

    V.         Typo mistakes are there in the document.

  VI.         Some references are not in the proper format.

VII.         Please elaborate conclusion by adding limitations as well as future work of the proposed study and must be relevant to the proposed research.

Author Response

Point 1: Please separately add problem statement and contribution of the study in Introduction section.  

Response 1: We have added the relevant content to lines 105-110 of our new manuscript.

Point 2: Mention the challenges faced while implementation and how did you tackle these.

Response 2: The main challenge we faced was the parameter tuning of the model and we use grid search to solve this problem. We have added the relevant content to lines 200-201 of our new manuscript.

Point 3: Mention if there are any trade-offs for implementation of this approach while we deviate from traditional practices.

Response 3: We thank you for raising this comment. Our study is more of an extension of the traditional study than an offset to the traditional study. We have added the relationship between traditional study and our study to lines 110-113 of our new manuscript.

Point 4: Some punctuation errors are observed. Please improve the punctuation in the manuscript.

Response 4: Thank you very much for your careful observation. We have improved them.

Point 5: Typo mistakes are there in the document.

Response 5: Thank you very much for your careful observation. We have improved them.

Point 6: Some references are not in the proper format.

Response 6: Thank you very much for your careful observation. We have improved them.

Point 7: Please elaborate conclusion by adding limitations as well as future work of the proposed study and must be relevant to the proposed research.

Response 7: We have added the relevant content to lines 403-414 of our new manuscript.

Reviewer 4 Report

The manuscript examined risk factors of abnormal liver function in workers in the automotive manufacturing industry based on data in a certain region--Chongqing, China. Statistical analyses (e.g., hypothesis tests, logistic regression) were used to identify significant factors, then the deep neural network (DNN) was used to build a predictive model for future prediction tasks. The writing of the manuscript is of high quality. Modeling and results are comprehensive. However, there are some major limitations that require further discussion in the manuscript:

1. The key findings that "increases BMI, length of service, DBP, SBP, and being male are independent risk factors for abnormal liver function" are not very exciting to readers as they are more like common sense. However, I understand that the predictive model that takes advantage of these factors might be the emphasis of the manuscript. The weakness of these key factors still needs to be explicitly discussed.

2. The identified key factors may be useful to a wide range of researchers. However, the predictive model, based on data from Chongqing, can only be used to predict for the same population. This is a major limitation of the manuscript. As this is an international journal not a Chongqing-city journal, authors should discuss how this predictive model can be generally useful.

3. DNN is a kind of black-box modeling. The model is not as explicit as the typical regression models or tree-based models, where you can provide an equation or rules for future predictions. The re-use of the DNN model will be a big limitation. I didn't see how readers could take advantage of the DNN model in the future. And of course, the model itself was not presented in this manuscript as it's impossible to describe. Probably an open-source package/program/tool needs to be published together with this manuscript.

Author Response

Point 1: The key findings that "increases BMI, length of service, DBP, SBP, and being male are independent risk factors for abnormal liver function" are not very exciting to readers as they are more like common sense. However, I understand that the predictive model that takes advantage of these factors might be the emphasis of the manuscript. The weakness of these key factors still needs to be explicitly discussed.

Response 1: We thank you for raising this comment. We have discussed the relevant content in the section discussion of our new manuscript (lines 350-354).

Point 2: The identified key factors may be useful to a wide range of researchers. However, the predictive model, based on data from Chongqing, can only be used to predict for the same population. This is a major limitation of the manuscript. As this is an international journal not a Chongqing-city journal, authors should discuss how this predictive model can be generally useful.

Response 2: As you say, the limitation of the study population to Chongqing is an important limitation of this study. We add it as a limitation of the study in lines 406 to 407 of our new manuscript. To initially address this issue, we have uploaded the code and trained models for this study to the github repository. We hope that there are researchers around the world who can use their data for further validation of our study.

Point 3: DNN is a kind of black-box modeling. The model is not as explicit as the typical regression models or tree-based models, where you can provide an equation or rules for future predictions. The re-use of the DNN model will be a big limitation. I didn't see how readers could take advantage of the DNN model in the future. And of course, the model itself was not presented in this manuscript as it's impossible to describe. Probably an open-source package/program/tool needs to be published together with this manuscript.

Response 3: We saved our trained model as an R file and uploaded it to the github repository. Anyone who needs it can find my model and use it in real work.

Reviewer 5 Report

This study proposes a model for classifying normal/abnormal liver function in workers in the automotive manufacturing industry.

Suggestions and questions (answers could/should be used to improve the manuscript):

1. All abbreviations should be declared (e.g., BMI, DBP, SBP

2. Consider "The model based on the deep neural network has excellent performance and can be used as an effective tool to predict workers suffering from abnormal liver function and provide personalized management plans.",

3. The objective of the study should be declared at the end of the introduction.

4. The methodology explaining the data set and the use of machine/deep learning is extremely poor.

-Some data samples could be provided for illustration.

-Why was cross validation not used?

-Why was the ratio of 7:3 used?

-What was the evaluation metric used? ROC curve is not a metric. Consider that you have a binary classification problem.

-What was the DNN architecture used? Justify.

-What was the baseline model?

-How many interactions were there to train DNN?

-Was fine-tuning performed? What type? Grid search?

Moreover, the content about machine/deep learning should be in a section with heading "Statistical analysis".

5. Table 3 is poorly explained. It should be detailed.

6. Consider the text "...the final model hyperparameters are set as follows: the model contains 3 hidden layers, and the number of neurons in each hidden layer is 3.", why do you consider your model as 'deep"?

7. Consider "The model achieved good predictions...", why do you consider an accuracy of ~0.68 as 'good'? In the conclusion section, it is considered as one with 'excellent performance'.

8. Do you consider your study using 'deep' learning with no limitations? None were acknowledged in the discussion.

9. Finally, where are the related works? A comparison with previous studies should be added.

Specific comments:

- e.g. -> e.g., (add comma)

- et al -> et. al (add dot)

- line 69: "...function. And a timely..." - avoid to start a sentence with "And".

- line 117: "All case physical examination data for workers are obtained from occupational health..." -> were? past

- line 121: Avoid "etc"

- Check "Table 2. Comparison of characteristics between normal liver function group and liver function group." - ABNORMAL liver function group?

Author Response

Point 1: All abbreviations should be declared (e.g., BMI, DBP, SBP  

Response 1: We thank you for raising this comment. We have added it to line 415 of our nuw manuscript.

Point 2: Consider "The model based on the deep neural network has excellent performance and can be used as an effective tool to predict workers suffering from abnormal liver function and provide personalized management plans."

Response 2: Thank you for your valuable comment. This sentence was really not well phrased and we have removed it.

Point 3: The objective of the study should be declared at the end of the introduction.

Response 3: We have added the relevant content to lines 106-107 of our new manuscript.

Point 4: The methodology explaining the data set and the use of machine/deep learning is extremely poor.

-Some data samples could be provided for illustration.

-Why was cross validation not used?

-Why was the ratio of 7:3 used?

-What was the evaluation metric used? ROC curve is not a metric. Consider that you have a binary classification problem.

-What was the DNN architecture used? Justify.

-What was the baseline model?

-Was fine-tuning performed? What type? Grid search?

Moreover, the content about machine/deep learning should be in a section with heading "Statistical analysis".

Response 4: We have improved the relevant parts of the data set and the use of machine/deep learning in the "Statistical analysis" section.

-Some data samples could be provided for illustration.

We have showned some data samples in Table 2.

-Why was cross validation not used?

We used cross-validation in helping us to tune the model parameters, and for subsequent model evaluation we used a method that divides the train and test sets.

-Why was the ratio of 7:3 used?

Because our sample has a total of 6087 cases, and our model needs a larger sample size for training to achieve good prediction results. A certain number of samples are also needed for testing to evaluate the generalization ability of the model. In order to achieve good results for both, we chose a ratio of 7:3 to divide the training and testing sets.

-What was the evaluation metric used? ROC curve is not a metric. Consider that you have a binary classification problem.

We built three other models and evaluated the model performance using five metrics: AUC, accuracy, sensitivity, specificity, and F1-score. We have added the relevant content to lines 195-199 of our new manuscript and added a new figure (Figure 4.) to compare the four models.

-What was the DNN architecture used? Justify.

In this study we used a fully-connected neural network architecture to train our DNN model. We have added the relevant content to lines 188-189 of our new manuscript.

-What was the baseline model?

We built three other common models as the baseline models.

-Was fine-tuning performed? What type? Grid search?

We performed fine-tuning using grid search. We have added the relevant content to lines 200-201 of our new manuscript.

Point 5: Table 3 is poorly explained. It should be detailed.

Response 5: To make our presentation clearer, we have reorganized the language to illustrate the contents of Table 3 (lines 237-245).

Point 6: Consider the text "...the final model hyperparameters are set as follows: the model contains 3 hidden layers, and the number of neurons in each hidden layer is 3.", why do you consider your model as 'deep"?

Response 6: We are sorry that our vague expressions have brought you this doubt. In some books and literatures, we see that a neural network with hidden layers greater than 1 can be called a deep neural network. The neural network we constructed has 3 hidden layers, more than 1 layer, and can be called a deep neural network. We used “deep” here not to describe the deep depth of the network we are building (many hidden layers and neurons), but rather a way to name our network (to distinguish it from a single hidden layer neural network).

Point 7: Consider "The model achieved good predictions...", why do you consider an accuracy of ~0.68 as 'good'? In the conclusion section, it is considered as one with 'excellent performance'.

Response 7: We use the word “good” because the AUC value of 0.764 and the accuracy of 0.689 are relatively good compared to a large number of other medically relevant modeling studies (such as AUC value of “Wang Y, Wang L, Su Y, Zhong L, Peng B. Prediction model for the onset risk of impaired fasting glucose: a 10-year longitudinal retrospective cohort health check-up study”). The expression 'excellent performance' is indeed a misuse of our words and has been revised.

Point 8: Do you consider your study using 'deep' learning with no limitations? None were acknowledged in the discussion.

Response 8: Please forgive our lack of knowledge and shallow understanding of deep learning. We use the term “deep learning” in our research because we learned about DNN networks in a book called "Deep Learning and the R Language". In this book, the authors build neural networks with sometimes as few as two hidden layers, and the authors still call such networks deep learning. We further searched the definition of deep learning in Wikipedia and found the statement "There is no universally agreed depth threshold to classify shallow learning and deep learning. However, most researchers agree that deep learning involves a CAP depth higher than 2". And we have seen in other sources that neural networks with more than one hidden layer can be called deep learning. We have also tried to increase the number of hidden layers of our neural network to make it deeper in our research, but we found that overfitting occurs at more than 3 hidden layers in our data. We finally chose to build our neural network with only 3 hidden layers. We have added relevant acknowledgments in lines 373-377 of our new manuscript.

Point 9: Finally, where are the related works? A comparison with previous studies should be added.

Response 9: We have added relevant content in lines 87-102 of our new manuscript.

Point 10: Specific comments:

- e.g. -> e.g., (add comma)

- et al -> et. al (add dot)

- line 69: "...function. And a timely..." - avoid to start a sentence with "And".

- line 117: "All case physical examination data for workers are obtained from occupational health..." -> were? past

- line 121: Avoid "etc"

- Check "Table 2. Comparison of characteristics between normal liver function group and liver function group." - ABNORMAL liver function group?

Response 10: We appreciate your careful review of our manuscript and your valuable comments. We apologize for making these mistakes due to our oversight, and we have corrected them in our new manuscript.

Reviewer 6 Report

Novelty is unclear.

 (-) The introduction must be improved.

 (-) The related work section must be enhanced.

 (-) The method is too simple.

 (-) The method is not novel enough.

 (-) Experimental evaluation must be improved.

 (-) Some improvements are needed in the description of the method.

The authors need to better explain the context of this research, including why the research problem is important.

The introduction should clearly explain the key limitations of prior work that are relevant to this paper.

Contributions should be highlighted more. It should be made clear what is novel and how it addresses the limitations of prior work. 

The authors should explain clearly what  the differences are between the prior work and the solution presented in this paper.

The related work section is too short.

The authors should add a table that compares the key characteristics of prior work to highlight their differences and limitations. The authors may also consider adding a line in the table to describe the proposed solution.

A novel solution is presented but it is important to better explain the design decisions (e.g. why the solution is designed like that)

It is important to clearly explain what is new and what is not in the proposed solution. If some parts are identical, they should be appropriately cited and differences should be highlighted.

The solution is described but there should be more examples.

The description of the proposed solution should be more formal.

 The authors should add proof(s) of the properties, theorem or lemmas contained in the paper.

It is necessary to discuss the complexity of the proposed solution.

The experiments should be updated to include some comparison with newer studies. 

A statistical analysis should be carried out to demonstrate that the experimental results are significant. 

There is not enough discussion of the experimental results. 

The experiments have been carried with a few datasets. It is necessary to add more datasets so as to make experiments more convincing.

Some text must be added to discuss the future work or research opportunities.

Author Response

Point 1: Novelty is unclear.  

Response 1: The novelties of our study are the study population (auto manufacturing workers) and the variables (demographic characteristics and occupational information) used to develop model. We have added the relevant content to lines 104-110 of our new manuscript.

Point 2: The introduction must be improved.

Response 2: We have improved the introduction of our manuscript.

Point 3: The related work section must be enhanced.

Response 3: We have added the relevant content to lines 87-103 of our new manuscript.

Point 4: The method is too simple.

Response 4: We have improved our method mainly includes the evaluation part of the model.

Point 5: The method is not novel enough.

Response 5: The novelties of our study are the study population (auto manufacturing workers) and the variables (demographic characteristics and occupational information) used to develop model. We have added the relevant content to lines 104-110 of our new manuscript.

Point 6: Experimental evaluation must be improved.

Response 6: We built three other models and evaluated the model performance using five metrics: AUC, accuracy, sensitivity, specificity, and F1-score. We have added the relevant content to lines 195-199 of our new manuscript and added a new figure (Figure 4.) to compare the four models.

Point 7: Some improvements are needed in the description of the method.

Response 7: We have made reasonable improvements to the description of the method section, mainly including a detailed description of the model architecture, the selection of model parameters, and the performance evaluation of the model.

Point 8: The authors need to better explain the context of this research, including why the research problem is important. Contributions should be highlighted more. It should be made clear what is novel and how it addresses the limitations of prior work. 

Response 8: Thank you very much for your valuable comment. We have improved our introduction of our manuscript. (lines 105-113)

Point 9: The introduction should clearly explain the key limitations of prior work that are relevant to this paper. The related work section is too short. The authors should add a table that compares the key characteristics of prior work to highlight their differences and limitations. The authors may also consider adding a line in the table to describe the proposed solution. The authors should explain clearly what the differences are between the prior work and the solution presented in this paper. It is important to clearly explain what is new and what is not in the proposed solution. If some parts are identical, they should be appropriately cited and differences should be highlighted.

Response 9: Your comments has benefited me greatly. Based on your suggestions, I summarized the characteristics and flaws (Table 1.) of previous research and how my research was conducted to address these flaws. These content changes are in lines 87-103 of our manuscript.

Point 10: There is not enough discussion of the experimental results. 

Response 10: Based on your suggestions, we have enriched the discussion of our results (lines 370-396).

Point 11: The experiments have been carried with a few datasets. It is necessary to add more datasets so as to make experiments more convincing.

Response 11: This is the main limitation of this research. We have added it in lines 406-407 of our new manuscript.

Point 12: Some text must be added to discuss the future work or research opportunities.

Response 12: We have added relevant content in lines 407-414 of our new manuscript.

Round 2

Reviewer 4 Report

I see significant improvement in this revision. Though this study still has flaws, especially the generalizability issue of the predictive model, this has been explicitly discussed in the limitation of the manuscript. The new version is worth publishing for further discussions in the research community.

Author Response

Point 1: I see significant improvement in this revision. Though this study still has flaws, especially the generalizability issue of the predictive model, this has been explicitly discussed in the limitation of the manuscript. The new version is worth publishing for further discussions in the research community.

Response 1: Thank you again for your valuable comments on our manuscript, which have been of great value to us in improving our manuscript.

Reviewer 5 Report

The authors have improved the paper, and addressed some of my comments. Some of them require additional attention:

Point 4: The methodology explaining the data set and the use of machine/deep learning is extremely poor.

-Some data samples could be provided for illustration.

-Why was cross validation not used?

-Why was the ratio of 7:3 used?

-What was the evaluation metric used? ROC curve is not a metric. Consider that you have a binary classification problem.

-What was the DNN architecture used? Justify.

-What was the baseline model?

-Was fine-tuning performed? What type? Grid search?

Moreover, the content about machine/deep learning should be in a section with heading "Statistical analysis".

Response 4: We have improved the relevant parts of the data set and the use of machine/deep learning in the "Statistical analysis" section.

-Some data samples could be provided for illustration.

We have showned some data samples in Table 2.

-Why was cross validation not used?

We used cross-validation in helping us to tune the model parameters, and for subsequent model evaluation we used a method that divides the train and test sets.

-Why was the ratio of 7:3 used?

Because our sample has a total of 6087 cases, and our model needs a larger sample size for training to achieve good prediction results. A certain number of samples are also needed for testing to evaluate the generalization ability of the model. In order to achieve good results for both, we chose a ratio of 7:3 to divide the training and testing sets.

-What was the evaluation metric used? ROC curve is not a metric. Consider that you have a binary classification problem.

We built three other models and evaluated the model performance using five metrics: AUC, accuracy, sensitivity, specificity, and F1-score. We have added the relevant content to lines 195-199 of our new manuscript and added a new figure (Figure 4.) to compare the four models.

-What was the DNN architecture used? Justify.

In this study we used a fully-connected neural network architecture to train our DNN model. We have added the relevant content to lines 188-189 of our new manuscript.

-What was the baseline model?

We built three other common models as the baseline models.

-Was fine-tuning performed? What type? Grid search?

We performed fine-tuning using grid search. We have added the relevant content to lines 200-201 of our new manuscript.

Reviewer's reply:

1. Consider the sentences "All samples were divided into training and test sets in the ratio of 7:3. The training set was used to train our model. We used grid search and 10-fold cross-validation for model parameter tuning". If you have used 10-fold cross-validation, so your ratio is 9:1, not 7:3.

2. If you have used Grid search for fine-tuning, what are the best hyperparameters? Consider all of them used in the grid search, including epochs.

3. I mean, "the content about machine/deep learning should 'NOT' be in a section with heading Statistical analysis". That is not statistical analysis.

Point 7: Consider "The model achieved good predictions...", why do you consider an accuracy of ~0.68 as 'good'? In the conclusion section, it is considered as one with 'excellent performance'.

Response 7: We use the word “good” because the AUC value of 0.764 and the accuracy of 0.689 are relatively good compared to a large number of other medically relevant modeling studies (such as AUC value of “Wang Y, Wang L, Su Y, Zhong L, Peng B. Prediction model for the onset risk of impaired fasting glucose: a 10-year longitudinal retrospective cohort health check-up study”). The expression 'excellent performance' is indeed a misuse of our words and has been revised.

Reviewer's reply:

4. You should explain in the manuscript why you consider 'good' performance. This requires a justification.

Point 8: Do you consider your study using 'deep' learning with no limitations? None were acknowledged in the discussion.

Response 8: Please forgive our lack of knowledge and shallow understanding of deep learning. We use the term “deep learning” in our research because we learned about DNN networks in a book called "Deep Learning and the R Language". In this book, the authors build neural networks with sometimes as few as two hidden layers, and the authors still call such networks deep learning. We further searched the definition of deep learning in Wikipedia and found the statement "There is no universally agreed depth threshold to classify shallow learning and deep learning. However, most researchers agree that deep learning involves a CAP depth higher than 2". And we have seen in other sources that neural networks with more than one hidden layer can be called deep learning. We have also tried to increase the number of hidden layers of our neural network to make it deeper in our research, but we found that overfitting occurs at more than 3 hidden layers in our data. We finally chose to build our neural network with only 3 hidden layers. We have added relevant acknowledgments in lines 373-377 of our new manuscript.

Reviewer's reply:

5. Is this justification provided in the manuscript? If not, provide it.

6. In addition, the subsection "Study limitations and future works" should be in the discussion section.

Author Response

Point 1: Consider the sentences "All samples were divided into training and test sets in the ratio of 7:3. The training set was used to train our model. We used grid search and 10-fold cross-validation for model parameter tuning". If you have used 10-fold cross-validation, so your ratio is 9:1, not 7:3.

Response 1: Thank you again for your careful reading of our manuscript and for your very valuable comments, which have benefited us greatly. We are sorry for the misunderstanding caused by our vague statement. We first divided all samples into a train set and a test set in the ratio of 7:3. The train set is used to train the model and parameter tuning, and the test set is used to evaluate the generalization ability of the models. No parameter tuning is required for the LR model. In the training set, we used a grid search in order to perform parameter tuning while training the three models. A 10-fold cross-validation method (in the train set) was used to select the optimal parameters. After selecting the optimal parameters the final models used are determined, and they are used to make predictions on the test set data and to derive the final evaluation metrics. Overall, we used both a 7:3 ratio to divide the training and test sets (validation of the generalization ability of the model) and a 10-fold cross-validation to train our model (parameter tuning of the model). We have added a more detailed formulation to lines 201 to 206 of our new manuscript.

Point 2: If you have used Grid search for fine-tuning, what are the best hyperparameters? Consider all of them used in the grid search, including epochs.

Response 2: Thank you for this comment. We have added the optimal parameters of the model as supplementary material (Table S1.) to our new manuscript (lines 277-278). About the epoch, we have added it to our new manuscript (lines 281-282).

Point 3: I mean, "the content about machine/deep learning should 'NOT' be in a section with heading Statistical analysis". That is not statistical analysis.

Response 3: We apologize that we misinterpreted your meaning and have changed that section to a new section in our new manuscript (Section 2.4. Development and evaluation of DNN model).

Point 4: You should explain in the manuscript why you consider 'good' performance. This requires a justification.

Response 4: We have added the related content to our new manuscript (lines 398-402).

Point 5: Is this justification provided in the manuscript? If not, provide it.

Response 5: We have added the related content to our new manuscript (lines 379-388).

Point 6: In addition, the subsection "Study limitations and future works" should be in the discussion section.

Response 6: Thank you very much for your attentive review of our manuscript. We have changed the subsection "Study limitations and future works" to the discussion section.

Reviewer 6 Report

The related work still needs to be enhanced.

 The method is not novel enough.

  Improvements are needed in the description of the method.

The authors failed to explain clearly what  the differences are between the prior work and the solution presented in this paper.

The claim is novel solution is presented but it is important to better explain the design decisions (e.g. why the solution is designed like that)

Author Response

Point 1: The related work still needs to be enhanced.

Response 1: We have added the relevant content to lines 87-103 of our new manuscript. A table (Table 1.) is summarized to compare the difference between the related work and our study.

Point 2: The method is not novel enough.

Response 2: The novelties of our study are the study population (auto manufacturing workers) and the variables (demographic characteristics and occupational information) used to develop model. The DNN model was first used in auto manufacturing workers. We have added the relevant content to lines 104-110 of our new manuscript.

Point 3: Improvements are needed in the description of the method.

Response 3: We have made reasonable improvements to the description of the method section, mainly including a detailed description of the model architecture, the selection of model parameters, and the performance evaluation of the model.

Point 4: The authors should explain clearly what the differences are between the prior work and the solution presented in this paper.

Response 4: Your comment has benefited me greatly. Based on your suggestions, I summarized the characteristics and flaws (Table 1.) of previous research and how my research was conducted to address these flaws. These content changes are in lines 87-103 of our manuscript.

Point 5: The claim is novel solution is presented but it is important to better explain the design decisions (e.g. why the solution is designed like that) 

Response 5: We are sorry that our vague expressions caused you to misunderstand our research. We did not claim a novel solution in the study. In fact, the main innovation of this study is that the the study population (auto manufacturing workers) and the variables (demographic characteristics and occupational information) used to develop model.
